# Home, sweet home? The impact of working from home on the division of unpaid work during the COVID-19 lockdown

Judith Derndorfer[1]☯, Franziska Disslbacher[1,2,3]☯*, Vanessa Lechinger[1]☯*, Katharina Mader[2,4]☯, Eva Six[1]☯

**1** Research Institute Economics of Inequality, WU Vienna University of Economics and Business, Vienna, Austria, **2** Department of Economics and Statistics, Chamber of Labor, Vienna, Austria, **3** Institute for Economic Geography & GIScience, WU Vienna University of Economics and Business, Vienna, Austria, **4** Department of Economics, WU Vienna University of Economics and Business, Vienna, Austria

☯ These authors contributed equally to this work.
* franziska.disslbacher@wu.ac.at (FD); vanessa.lechinger@wu.ac.at (VL)

**Data Availability Statement:** All datafilee are available from the OSF database (accession link https://osf.io/sr7yu/).

## Abstract

A lockdown implies a shift from the public to the private sphere, and from market to non-market production, thereby increasing the volume of unpaid work. Already before the pandemic, unpaid work was disproportionately borne by women. This paper studies the effect of working from home for pay (WFH), due to a lockdown, on the change in the division of housework and childcare within couple households. While previous studies on the effect of WFH on the reconciliation of work and family life and the division of labour within the household suffered from selection bias, we are able to identify this effect by drawing upon the shock of the first COVID-19 lockdown in Austria. The corresponding legal measures left little choice over WFH. In any case, WFH is exogenous, conditional on a small set of individual and household characteristics we control for. We employ data from a survey on the gendered aspects of the lockdown. The dataset includes detailed information on time use during the lockdown and on the quality and experience of WFH. Uniquely, this survey data also includes information on the division, and not only magnitude, of unpaid work within households. Austria is an interesting case in this respect as it is characterized by very conservative gender norms. The results reveal that the probability of men taking on a larger share of housework increases if men are WFH alone or together with their female partner. By contrast, the involvement of men in childcare increased only in the event that the female partner was not able to WFH. Overall, the burden of childcare, and particularly homeschooling, was disproportionately borne by women.

## 1 Introduction

Crises and measures to cope with them exert a different impact on men and women, regardless of whether the nature of the crisis is economic (e.g. [1, 2]), environmental (e.g. [3–5]) or social (e.g. [6, 7]). The COVID-19 pandemic is no exception to this rule. Following the rapid spread

**Funding:** KM received funding from the Vienna Science and Technology Fund (WWTF), grant number COV20-040 (URL of the funder: https://www.wwtf.at). KM received funding from the Vienna Chamber of Labour under the grant "Multiple Burdens of COVID-19" (URL of the funder: https://wien.arbeiterkammer.at/index.html) VL, JD and ES were funded by these grants. The funders did not play any role in the study design, data collection and analysis, decision to publish, or preparation of the manuscript.

**Competing interests:** The authors have declared that no competing interests exist.

of the SARS-CoV-2 virus in early 2020, the immediate response of governments across the world was to lockdown large parts of the economy to slow down the spread of the virus and to mitigate negative effects on public health. A tremendous amount of research investigates the impact of the pandemic and the measures implemented to cope with it on social and economic outcomes. For instance, as of March 10 2021, the National Bureau of Economic Research has published 375 working papers on COVID-19. The vast majority of that research studies aspects that are subject to official statistics, particularly labour market statistics and GDP, and stresses the gendered, but country-specific, effects of the pandemic. The gendered labour market effects of lockdowns, for instance, depend on the specific restrictions in relation to the sectoral composition of the economy, and the sectoral composition by gender in particular. Early studies on the US, the UK, Australia and Spain have shown that women were disproportionately affected by job loss, short-time work and reduction of working time [8–12] in the first months of the pandemic. However, in Austria and Germany unemployment and short-time work hit women and men to a similar extent in the first weeks of the pandemic (March to May 2020), while male employment recovered quicker than female employment in the second half of the year [13, 14]. Yet even in the latter countries, gendered effects of the pandemic were noticeable early on: studies for Germany [14] have shown that the reduction in minor employment, which is not covered by unemployment insurance and short-time work schemes, was disproportionately notable for women. However, spheres not subject to official or regular data production efforts usually remain a blind spot. This paper sheds light on one of these economically and socially significant blind spots: unpaid work. We study how the shift towards working from home (WFH) for pay due to the first, strict lockdown in Austria has had an impact on the division of unpaid work within households.

Unpaid work is conducted to provide unpaid domestic services for use within the household and for reproduction. It includes housework, care given to household members and others and the provision of community services [15]. Across the world, women work longer unpaid hours than men [15, 16] Based on a collection of 133 time use surveys carried out in 76 countries during the past 20 years, Charmes (2019) reports that globally "women carry out three-quarters of unpaid care work, or more than 75% of the total hours provided" [15, p. 3]. This most comprehensive study, in terms of the world's population covered, concludes that there is not a single country where women and men perform an equal share of unpaid care work. Across countries, women's share in unpaid work ranges from 55.3% in Sweden to 92% in Mali. It is only in three Scandinavian countries (Sweden, Norway and Denmark), where the female share of unpaid work is below 60%. According to that report, women conducted 64.8% of unpaid work in Austria in 2008/2009 which is a substantially higher fraction as compared to Scandinavia and other high-income countries, such as Canada. From a global perspective, Austria is located among the half of countries where the women's share of unpaid work relative to their total work share, paid and unpaid, is particularly high (62.7% of total work). These numbers are roughly in line with a report of Statistics Austria that is based on the same data, namely the latest time use survey conducted in Austria in 2008/2009 [17]. It finds that women in Austria do two thirds of unpaid work and one third of paid work.

Numerous approaches provide an explanation of the gendered patterns of time use, ranging from time availability approaches [18], over bargaining and separate spheres perspectives [19, 20], to the gender display approach [21–23]. Most of these theories stress that the division of labour within the household results from gendered power relations, which in turn are due to various factors: "some quantifiable, such as individual economic assets, others less so, such as communal/external support systems or social norms and institutions, or perceptions about contributions and needs" [24, p. 7]. Importantly, unpaid work enables productive and paid economic activity and stabilizes the economy in times of crisis. Despite its pivotal role for the

economy, unpaid work is not counted as productive work in conventional productivity measures or GDP. Moreover, as unpaid work is invisible, it remains unrecognized in most policy decisions and is frequently neglected due to the belief that what happens in the household is a private matter. This became evident during the pandemic, as governments closed kindergartens and schools, while taking the provision of unpaid work in the home for granted [25].

In the economic literature, the notion of asymmetric intra-household bargaining as a mechanism that causes an unequal distribution of both paid and unpaid work among household members appeared first in the bargaining models of Manser and Brown (1980) [26] and McElroy and Horney (1981) [27]. Bargaining models describe the intra-household allocation of resources as an outcome of a rational bargaining processes, and the individual household members are recognized as separate agents, each having their own preferences and a distinct utility function. The many models that have been suggested differ in their assumptions on the determinants of individual bargaining power, but typically, the access to economic resources, such as earnings or wealth, is emphasized as a critical determinant of an individual's degree of power [20, 28–30]. Evidence on the resources subject to bargaining and the consequences of intra-household decision making exists primarily for countries located in the Global South [31, 32], for instance, in terms of decisions over health and nutrition [33–35]. For European countries, there is less empirical evidence on intra-household dynamics and the distribution of decision-making power within households. However, Dema-Moreno (2009) [36] studies the decision-making processes of Spanish couples, while Lyngstad et al. (2011) [37] focus on Norwegian couples, Mader et al. (2012) [38] investigate the gendered distribution of income and decision-making power in Austrian households, and Sikorski and Kuchler (2012) [39] study decision-making in German households. Note that an important caveat of the present study is that we investigate and observe the outcome of bargaining and decision making processes, in contrast to studies on the process itself (for example [38]). The aforementioned empirical studies reveal that decisions on the allocation of time and money within the household are frequently made spontaneously, result from established practice, or have an outcome that conforms to social norms while support for actual and rational bargaining is limited. Overall, these findings suggest that women and men take on different tasks based on prevailing gender roles and gendered attributions, and that social norms encourage women to assume the primary care-taking role and to conduct the bulk of unpaid housework.

The COVID-19 lockdowns caused a substantial increase in the volume of unpaid work by shifting production from paid to unpaid work and thus from market to non-market production. This particularly affected parents of young children. The closure of restaurants, canteens and bars translates into more time spent on grocery shopping and the preparation of meals at home. The lockdown of childcare institutions and schools increases the volume of unpaid work by shifting care almost exclusively to the home. This is intensified by contact restrictions that make cleaning staff and nannies employed by households unavailable. The hours spent on unpaid work also increased as the support of grandparents, relatives and friends was to be avoided in order protect their health and save lives. This overall increase in the volume of hours spent on unpaid childcare during lockdown is documented in a number of studies, and estimates range from an increase of 25% in Spain [8], to 37% in Hungary [40], up to double the pre-lockdown hours in the United Kingdom [41]. These and additional studies also find that during COVID-19 lockdowns, women worked longer hours unpaid than men [42, 43].

In addition to causing a gender-specific increase in the volume of unpaid work, lockdowns also imply a shift towards work from home for pay (WFH) for non-essential these are individuals working outside *essential* sectors, such as grocery stories and healthcare that were not shut down during the pandemic. Importantly, the definition of "essential", and correspondingly, of "non-essential", varies by country, in the sense that the sectors regarded as "essential" and

unaffected by lockdown measures differ across countries. workers. As a result, many individuals and households have to rearrange their entire (paid and unpaid) work life. Thus, as the lockdown shifts the locus of production to the home, the household becomes the prime location of both market and non-market production. Thereby the barriers between WFH and unpaid work, between the public (paid) and the private (unpaid) sphere are blurred. Such a shift towards the home and household production also provokes behavioural responses that feed back into the public sphere and the economy. For instance, to cope with the increased volume of unpaid work, in particular childcare, mothers were more likely than fathers to reduce paid working hours in response to the lockdown [43–45]. Hence, the public and the private, the notion and extent of paid and unpaid work, are by no means separate spheres of work; rather—and this has clearly been revealed by COVID-19 lockdowns—they are interwoven and inseparable.

In this paper, we study the effect of WFH during the first, strict COVID-19 lockdown in Austria on the division of unpaid work within heterosexual couple households and the working conditions of WFH. From a conceptual point of view, we describe paid and unpaid work as interwoven dimensions of work and we answer three related research questions: did the involvement of males in housework and childcare increase during the lockdown as compared to before? Is there a gender gradient in the experience of WFH? What is the effect of WFH on the intra-household division of unpaid work? Our empirical strategy for estimating the effect of WFH on the change in the division of housework and childcare exploits the experimental setting provided by the lockdown measures. In essence, the pandemic and the following first lockdown are shocks exogenous to the demand of unpaid work. In Austria, the case studied in this paper, the design of the lockdown measures allowed for few possibilities to opt in and out of WFH. This fact makes it possible to identify the effect of lockdown-induced WFH on the intra-household division of unpaid work. We employ data collected from the survey *Multiple Burdens under COVID-19* that we conducted between April and May 2020, that is, during the first strict lockdown. Due to its relatively conservative views on gender roles, Austria is a country and case of great interest in this respect. According to the latest Eurobarometer survey No. 465 [46] almost 4 out of 10 residents agree that "the most important role of a woman is to take care of her home and family". Among EU-15 countries, the share of individuals who concurred with this statement is larger only in Portugal (47%), Italy (51%), Ireland (52%) and Greece (69%). By contrast, in countries which rank high on gender equality indices such as Sweden, Denmark and the Netherlands, the share of respondents agreeing with this statement is below 16%. In addition, Austria was, next to Italy, confronted with the rapid spread of the SARS-CoV-2 virus at an early stage of the pandemic. In the first weeks of the pandemic, Alon et al. (2020) [47] optimistically argued that the COVID-19 crisis would result in a more equal division of unpaid work within couple households, which would ultimately reduce gender inequality on the labour market. Thus, we test this assumption and examine whether WFH during the lockdown restrictions weakened or strengthened traditional gender roles as expressed in the division of housework and childcare.

We contribute to the literature on the gender-specific effects of lockdowns along the following lines. First, and most importantly, we present the first paper that studies the *change* in the division of unpaid work, that is to say, in the division of housework and childcare, within households due to a lockdown. While previous work on the gendered division of labour in high-income countries during lockdowns has focused on hours of unpaid work by gender, we investigate whether and to what extent the first, strict lockdown intensified the pre-lockdown gendered division of unpaid work within the household. Second, we focus on the interwoven situations of WFH and unpaid work and argue that the household composition of WFH is a central mechanism behind this change. Although household characteristics such as age of

household members, their education levels, and the hours worked for pay, for instance, are important determinants of the hours worked unpaid and the division of unpaid work within households, we expect to find a significant effect of WFH on the change in the within-household division of unpaid work conditional on these demographic and socio-economic characteristics. Third, we are able to identify the effect of WFH on the change in the division of unpaid work within households by exploiting the experiment provided by the first lockdown. While the impact of WFH on the division of unpaid work was already debated and studied before the COVID-19 pandemic (see for instance [48]), these contributions struggled to identify the effect of WFH, as in the investigated settings WFH could have been both a cause and a consequence of unpaid work.

## 2 Research design: Institutional setting, data and methods

In this chapter we introduce the research design. Specifically, we discuss the timing and nature of the lockdown measures, the data source, the sample definition and its characteristics, the definitions of the core variables, and the econometric strategy and estimation method we employ to study the effect of WFH due to the lockdown on the change in the division of unpaid work in couple households.

### 2.1 The first lockdown

The first COVID-19 patients were hospitalized as early as February 2020, yet it took a couple of weeks for the first legal measures to be announced and become effective in Austria. On 10th March 10 2020 the Austrian government announced the first regulations vastly restricting public and private life. Starting with Monday, 16th March, people could leave their homes only: (i) to attend their professional work if WFH was not feasible (such as for emergency services, the healthcare sector, or the food retail sector), (ii) to buy urgently needed goods (groceries, medicine, etc.), (iii) to look after care recipients or (iv) to exercise outside for one's physical and mental health. Thus, as of mid-March, restaurants, bars, hotels, nurseries, kindergartens, schools, universities, most offices, theaters, retail stores, other public institutions, and many more were temporarily closed. Only grocery stores, banks and pharmacies remained open. A couple of days later, an official obligation for "home office" the expression "home office" is used to describe WFH in Austria was announced a couple of days later, which reframed it as a 'target requirement', meaning that, if feasible, employers should let their employees WFH. The emergency ordinance BGBl. II Nr. 108/2020 declared that "professional activity should preferably take place outside the workplace". In practice, "home office" was enacted overnight for large parts of the working population, with no option to opt out. Prior to the lockdown working primarily from home was not widespread in Austria: before the outbreak of the pandemic, merely 2.6% of workers have been working primarily from home. However, there is substantial variation by employment status. While WFH was more common among self-employed (13.1%), it was a very rare practice among employees (1.1%) [49]. Additionally, with most of public life shut down, police enforced high fines whenever regulations were violated. This strict lockdown lasted one month, until after the Easter holidays (14th April), but "reopening" only started slowly on 1st May. While shops and stores could open again with strict safety measures on 14th April, childcare facilities and most educational institutions, businesses and food services, like restaurants, remained closed until the mid-Mid. Starting mid-May, schools opened and divided their students into alternating groups with each attending school only two days per week. With most offices staying closed and employees continuing to work from home, as well as most public childcare facilities still not fully operational, the "softer" lockdown period lasted until the end of June. In addition, the Austrian

government implemented a short-time work scheme. That is to say, while essential workers continued going to work and others WFH, a substantial share of the workforce was confronted with a drastic reduction in paid working hours.

## 2.2 Data and survey design

We use individual-level data from the cross-sectional survey *Multiple Burdens under COVID-19* that we conducted during the strict COVID-19 restrictions in Austria. The overall aim of this survey is to enable research on the gender-specific effects of the COVID-19 pandemic. Compared to related surveys conducted in other countries, the strength of this data is that it is both broad in scope and particularly detailed on the extent of unpaid work and its division within households before as well as during the lockdown. Applying the guidelines of Statistics Austria on time use surveys [17], respondents had to report their time use on the previous working day in intervals of 15 minutes for a set of given time use categories, and these intervals had to add up to 24 hours. This provided a detailed overview of how people spent their days during April and May 2020, a period characterized by limited possibilities for activities outside the home. However, we refrained from surveying time use before the lockdown in similar detail for several reasons. We refrained from surveying time use before the lockdown for the following reasons: first, several studies show that the respondents' memory of past events decreases with the time gap between the reference period and the timing of the interview, that is, recall bias increases. Reliable answers on pre-lockdown time use are unlikely, as their last working day was at least four to five weeks before the survey was released. Second, as filling out a time use model on a working day before the lockdown involves the provision of mental capacities and time of the respondents, we expected that the share of attrition, that is, the share of respondents not filling out the entire questionnaire, would be much higher in that case. In addition, it might have decreased the accuracy of answers to questions following the time use module substantially. Hence, the gains of the module might not outweigh the effort costs of the respondents finishing the survey properly. This implies that we are unable to compare the change in hours per activity (such as unpaid work tasks). However, we are able to study the change in the division of unpaid work within households by drawing on different questions.

In addition to information on time use, the data include rich information on the division, organization and quality of paid and unpaid work during the lockdown, on (satisfaction with) WFH, as well as a large set of socio-economic and demographic characteristics of the respondents and their partners and some information on any children who live in the same household. For standard items, such as the highest level education completed, the questionnaire was designed following other surveys, such as the European Survey on Income and Living Conditions 2020 [50], or—regarding time use—the last Austrian Time Use Survey of 2008/09 [17]. However, those questions that target information on WFH and the implications of the lockdown were adapted such that they could capture the novel situation of WFH. Unique features of the data are that they include information both on respondents' and their cohabiting partners' time use and on the division of unpaid work before as well as during the lockdown. We asked respondents living in couple households whether their partner was also willing to participate in the survey. If so, (a.i) they received an anonymous partner ID for their partner to enter, which enabled us to link their responses. If not, (a.ii) respondents had the opportunity to answer a "partner module" on the time use and key characteristics (such as age, gender, education) of their partner. Thus, while the sampling and (main) observational unit of the survey are individuals, we can depict household dynamics via this partner data.

We designed the questionnaire in the first weeks of the lockdown and implemented it by means of the software LimeSurvey is an online software tool for user-friendly implementation

of different types of online surveys. For more info please visit the LimeSurvey manual. Before starting to distribute the survey, it was extensively pre-tested. The sampling strategy, targeting respondents with and without children who worked from home, is best described as "limited snowball sampling": We distributed the survey via various mailing lists of the Vienna University of Economics and Business, the Vienna Chamber of Labour (that is, the legal representation of all dependent workers), and the Austrian transport and services union Vida. The call to answer the questionnaire was accompanied by the appeal to forward the survey to friends, family and colleagues. In addition, we posted the survey in groups of the social media platform Facebook these groups were selected on the basis that we expected a high share of WFH women to be members there. and on Twitter. The sampling strategy is hence a limited version of the standard snowball sampling design that exclusively samples based on the appeal to invite further respondents to answer a questionnaire. 2,081 respondents answered the entire survey between 20th April and 14th May 2020. As the snowball distribution strategy targeted individuals who were working from home at that time, the sample has a constraint: compared to the Austrian working population, it includes a disproportionately high share of individuals who completed tertiary education, who were obliged to WFH by the lockdown, and who live in Vienna, the capital city.

The present research conforms to the STROBE (Strengthening the reporting of observational studies in epidemiology) reporting guidelines for case-control and cross-sectional studies. The research underlying the manuscript was conducted at the WU Vienna University of Economics and Business (www.wu.ac.at) and it conforms to the *Directive of the Rectorate for Research at Vienna University of Economics and Business on Responsible Research and Scientific Integrity*, it adheres to the guides for good academic practice of the Austrian Agency for Scientific Integrity and hence was approved by the Vice-Rector for Research of WU Vienna. Participation in the survey was voluntary. Before starting to answer the survey respondents had to give written consent that the resulting data will be used—only—for scientific research purposes by researchers. Respondents were informed that data will be reported only such that the identification of individual respondents will not be possible. Survey respondents had not to report any personal information that would enable the identification of single respondents, and hence the data is fully anonymous.

## 2.3 Sample and key variables

The main interest of this paper is to study the effect of WFH on the change in the within-household division of unpaid work. For this reason, we restricted the overall sample to 558 heterosexual couples (1,116 individuals) who

b.i.  lived in the same household during the lockdown. We exclude all same-sex couples, as their number in the sample is too small to allow for valid statements.

b.ii.  were both either employed, self-employed or on short-time work at the time of the survey,

b.iii.  either one partner answered the "partner module" and thereby provided information on her/his partner, or both partners answered the survey and linked them via an anonymous partner ID, and

b.iv.  provided full information on our explanatory variables.

Restrictions (b.i) and (b.iii) are necessary for the study of intra-household dynamics, while (b.ii) reduces the sample to working couples. Restriction b.iv ensures a stable sample size for the descriptive as well as econometric analysis. The vast majority, 79.6%, of the observations

are from questionnaires answered by women providing information about themselves and their male partner. The remainder is from questionnaires answered by males. S1 Table in S1 File describes the characteristics of this sample in detail. In the following section, we discuss the main variables of interest and the covariates we include in the econometric analysis.

**2.3.1 Division of unpaid work within the couple household before and during the lockdown.** We measure the division of housework (HW) and childcare tasks (CC) before (HW.i and CC.i) and during (HW.ii and CC.ii) the lockdown separately. Respondents hence had to answer four distinct questions and rank their share of housework (HW.i and HW.ii) and childcare tasks (CC.i and CC.ii) on a scale from zero (indicating the "Woman does everything") to ten (indicating the "Man does everything") (see S2 Table in S1 File). In the middle (scale no. 5), the division of tasks is shared equally between partners. Housework includes cooking, shopping and cleaning, but also tasks like gardening, animal care or repair work. Childcare comprises basic care, teaching (homeschooling), and recreational activities like talking, reading or playing with a child. Respondents had to answer these questions after having reported their time use for each of these subcategories. Thus, we assume that they were aware of the definition of housework and childcare when answering these questions. We define two of the main variables of interest based on these questions about the division of unpaid work within the household. First, we define the two dependent variables for the econometric analysis. These are dummy variables indicating whether the male partner took on more housework or childcare tasks during the lockdown than before the lockdown. These dummies are defined by combining the responses to HW.i, HW.ii, CC.i and CC.ii with information on the gender of the respondent. These dummy variables equal to one, if the value on the corresponding 11-point scale (zero to ten) was reported as being at least one point higher during the lockdown than before. 28% of all couples indicated that the male partner took on at least marginally more HW and 34% of all couples with children reported an increased involvement of the male partner in CC (see S1 Table in S1 File). Thus, we define the *change* in the division of HW and CC as an increased involvement of the male partner in these tasks. Second, we employ responses to questions HW.i and CC.i as measures for the division of housework and childcare prior to the lockdown. For this purpose, we subdivide the two 11-scale variables (indicating the pre-lockdown division of HW and CC) into four categories: "Woman does much more" (scale nos. 0–2), "Woman does more" (scale nos. 3–4), "Equal" (scale no. 5), "Man does (much) more" (scale nos. 6–10). Owing to the fact that in very few households men are primarily responsible for housework and/or childcare, we did not differentiate between "more" and "much more" in the case of males (see S1 Table in S1 File).

**2.3.2 Working from home (WFH).** In order to measure WFH during the COVID-19 restrictions, respondents who stated they were currently employed, self-employed or in short-time work were asked if they do WFH entirely, partly respondents who worked partially from home were assigned to the WFH group as well. In a robustness check, we excluded respondents who were WFH partially (see S5 Table in S1 File), but the main regression results are not affected by this change in the sample or not at all. As we are interested in dynamics within couple households, we created a factor variable indicating whether within a heterosexual couple nobody, only the man, only the woman or both partners were WFH during the lockdown. S1 Table in S1 File shows that the majority of respondents were WFH during the COVID-19 lockdown: in 18% only the woman was WFH, in 8% of cases only the man was WFH, whereas in 64% of all couples both were WFH.

**2.3.3 Socio-economic characteristics (covariates).** In addition, we include several covariates measured at the household and individual level in the econometric model: the relative income of partners, the highest level of education completed, age, number and age of children living in the household, employment status and working hours (see S1 Table in S1 File). In

order to be able to assess the presence and extent of power dynamics as manifested in income differences between partners, respondents had to report their own net income in the last month, their total disposable household income and their partner's net income according to one of 15 income brackets. In the case of the respondent's own net income, for example, these brackets range from "less than €600" to "more than €8, 000". Based on this categorical income variable, we define a factor variable indicating whether a respondent earned either more, less, or roughly the same as their partner. We divided the sample into tertile categories of net income in order to differentiate between couples in which both partners earn roughly the same (i.e. the reference group), and couples in which either the female or the male partner has higher monthly earnings. In the econometric specification, we focus on couples with children younger than 15 years and we grouped children according to their age. The cutoffs between age groups reflect differences in the educational status of the children: 0–2 years (very young children), 3–5 years (kindergarten), 6–9 years (primary school) and 10–14 years (lower secondary school). S1 Table in S1 File shows the average number of children by age group and household type. Furthermore, we distinguish between individuals working part-time (less than 21 hours per week) and those who reduced their working time (involuntarily) due to short-time work. In addition, there is the group of "full-time short-time workers", which refers to respondents who were in short-time work but still worked more than 21 hours per week. The short-time work scheme in place while the survey was in the field allowed a reduction of working hours up to 90%.

## 2.4 Data analysis and econometric approach

In order to answer the research questions, we make use of descriptive statistics as well as standard econometric methods. In the first part, the descriptive analysis, we provide evidence on the gendered burden of WFH, characterizing the division of housework and childcare during the lockdown. In the second part, the econometric analysis, we study the effect of WFH on the change in the division of unpaid work within households. For this purpose, we estimate an econometric model that explains the probability that the male partner increased his share in unpaid work during the lockdown. Importantly, under "normal" circumstances, that is to say, without any COVID-19 restrictions in place, it is impossible to establish a clear relationship between (a shift towards) WFH and (the resulting change in) the division of unpaid work in an observational study. For instance, individuals can opt into or out of the treatment (WFH), resulting in an endogenous treatment and thus biased estimates. In that case, it would not be clear whether the option of flexible work, specifically WFH arrangements, is either a cause or a consequence of parents' involvement in household and care work. The COVID-19 restrictions in Austria offer an experimental setting that we can exploit to study the effect of WFH on the division of unpaid work: WFH was strongly recommended by regulation (see section 2.1) and enacted overnight. In practice, it was no longer a personal decision to WFH or not and there was no scope for planning. Within the population of working individuals, WFH can thus be considered as randomly assigned and exogenous. However, the legal WFH regulation entailed merely a strong recommendation to WFH. For this reason, we condition WFH on individual and household-level factors known to exert a key influence on the division of unpaid work as well as the pre-lockdown division of housework and childcare tasks. Thereby we are able to rule out any remaining possibilities for selection into or out of WFH.

We estimate a set of logistic regression models by maximum likelihood to investigate the effect of WFH on the binary dependent variable that describes changes in the division of unpaid work within couple households. Subsequently, we calculate the average of the corresponding sample marginal effects for each variable in the regression model. These average

marginal effects (AME) depict the average change in the probability that the dependent variable is true.

The population equation is given by Eq 1

$$Pr(Y_h = 1|z) = G(\beta_0 + \beta_1 WFH_{i,\ j\ in\ h} + \beta_2 D_{i,\ j\ in\ h} + \beta_3 X_{i,\ j\ in\ h} + \epsilon_h) \tag{1}$$

where $G(z)$ is the cumulative distribution function of the standard logistic distribution ($G(z) = \frac{exp(z)}{1+exp(z)}$), which, for all real numbers, takes on values strictly between zero and one ($0 < G(z)<1$). $\epsilon_h$ is assumed to have mean zero and a constant variance. Note the $h$ refers to the couple household $h = 1, \ldots, N$ nesting the partners $i, \ldots, I$ and $j, \ldots, J$. Thus, the dependent variable is the change in the division of housework at the household level, while the variables on the right are measured at the level of individuals nested in the corresponding household. For the sake of simplicity, we define $\beta = [\beta_0, \beta_1, \beta_2, \beta_2]$ as the vector of coefficients corresponding to the variables in the matrices $D_i$, $WFH_i$, and $X_i$, which together provide set of explanatory variables $z_i$ in the population equation. $\beta_0$ is a constant. We estimate Eq 1 by means of maximum likelihood. Hence, we define the density of $y_h$ conditional on $z_h$ as $f(y|z_h, \beta) = G(z_h \beta)^y[1 - G(z_h \beta)]^{1-y} \forall y = 0, 1$; the log-likelihood of an individual observation is given by $\ell_h(\beta) = y_j log (G(z_h \beta)) + (1 - y_h)[1 - G(z_h \beta)]$, while the likelihood function that is maximized in order to estimate $\beta$ is defined as $\mathcal{L}(\beta) = \sum_{h=1}^{n} \ell_h(\beta)$.

We work with two dependent variables, $Y$, thus, we estimate two variants of Eq 1. The first variable describes whether the male partner increased his share in housework (of the household) during the lockdown, the second depicts the increase in the male partner's involvement in childcare. The corresponding dummy variables are equal to one if the man proportionally took on more unpaid work during the lockdown, but it does not indicate how much more unpaid work this corresponds to than before the lockdown. Thus, the hours corresponding to the male part of the couple "doing more than before" can vary to a large extent in terms of hours. In other words, in this definition, every increase in the male partner's involvement in HW or CC counts equally, regardless of the corresponding hours. As we are interested in whether WFH changed the pre-lockdown division of unpaid work, conditional on covariates, we consider this to be the appropriate specification. The main explanatory variable hence is working from home (WFH), a factor variable measuring whether both, none, only the woman or only the man of the couple was WFH during the lockdown. In addition, we control for the division of unpaid work before the lockdown (D) and other household and individual characteristics (X), specifically relative income of partners, age, highest level of education completed, employment status, full-time or part-time work, and the number of children and age of any children living in the household.

## 3 Results

In this section we present the results of the descriptive and econometric analysis and we discuss the findings in more detail.

### 3.1 Descriptive results

This subsection presents the descriptive results of two overlapping spheres: the experience of WFH while coping with the increased demand for unpaid work (see section 3.1.1) and its division during the lockdown (see section 3.1.2). We reveal how the blurring of boundaries between the public and private domains and between work and family responsibilities have distinct implications on different types of households and genders.

**3.1.1 Working from home during lockdown restrictions.** As stated in chapter 2.1, the legal basis to WFH whenever it was *feasible* was enacted and communicated by the Austrian government starting in March 2020. Since most facilities, especially public spaces (like schools, universities, public buildings, libraries, restaurants, etc.) were closed, the term feasible was interpreted as *strongly advised* for workers in "non-critical infrastructure" (i.e. outside of supermarkets, elderly homes, hospitals, etc.). Especially with childcare facilities being closed and meeting up with friends and family being forbidden, Austrians spent most of their days at home. WFH during the lockdown therefore cannot be compared with WFH in non-pandemic and non-lockdown times. However, even by the end of 2020, no legal agreement on how WFH should be implemented was enforced in Austria. This implies that most employees had to manage WFH on their own from the beginning, but with few guidelines from their employers.

WFH entails both advantages and disadvantages. The most propagated benefits are not having to commute every day and an easier reconciliation of family and (paid) work. At the same time, contact with supervisors, managers or colleagues might be more limited. Another potential drawback is the blurring of boundaries between paid work and leisure time. In the survey, the respondents had to evaluate their current WFH situation by answering a set of questions. The corresponding questions covered different aspects, such as advantages and disadvantages of WFH, different forms of childcare when WFH and the quality of working time and their workspace at home. The respondents had to report how much they agreed or disagreed with statements on the quality of and their experience with WFH. They had to "Strongly agree", "Somewhat agree", "Somewhat disagree" or "Strongly disagree" with each of these statements. Only persons who worked either fully or at least partially from home answered these questions. In this section, we focus primarily on couple households with children under 15 years of age. The findings are shown in Fig 1, which captures the average agreement with different statements for fathers and mothers separately. The smaller the distance on the axis to the centre, the more the respondents disagree with the statement. To inverse the stereotypical colours of the sexes, blue areas represent answers from women and pink areas represent men's responses. The apparent gender-specific differences in couple households with children interestingly almost vanish in couple households without children (see S1 Fig in S1 File). Hence, the presence of children seems to make a vast difference in assessing the quality and (dis-)advantages of WFH.

Turning to the specific advantages and disadvantages of WFH, Fig 1A shows that mothers (blue area) in couple households with children found it more difficult to concentrate while WFH, to reconcile work and family life and to complete tasks better at home than at the office. Fathers (pink area) found these aspects easier than mothers, since their average agreement is higher. By contrast, communication with supervisors, the supervisor's recognition of their work performance, as well as contact with colleagues does not show any systematic relationship by gender. The results of the indicator on the quality of working time and of the workspace are shown in Fig 1B. The most striking gender differences concern the workspace and the separation of work from leisure time. Fewer mothers (blue area) agreed that they had their own room to work from, where they could close the door, compared to fathers (pink area). Furthermore, we find that, on average, slightly more mothers worked outside the agreed working hours: more mothers stated that they were accessible outside their agreed working hours, worked overtime and also at weekends.

Combining childcare and WFH during the lockdown was a difficult—and sometimes even unfeasible—task. Fig 2 displays different forms of childcare available while WFH. Among the respondents with children under 15 years old, 40.1% stated that the mother took care of the children during her (paid) working hours, supervising them in the same room. Contrastingly, only 24.8% reported that the father took care of the children while WFH. In almost 30% of the

## (A) Advantages and disadvantages of WFH

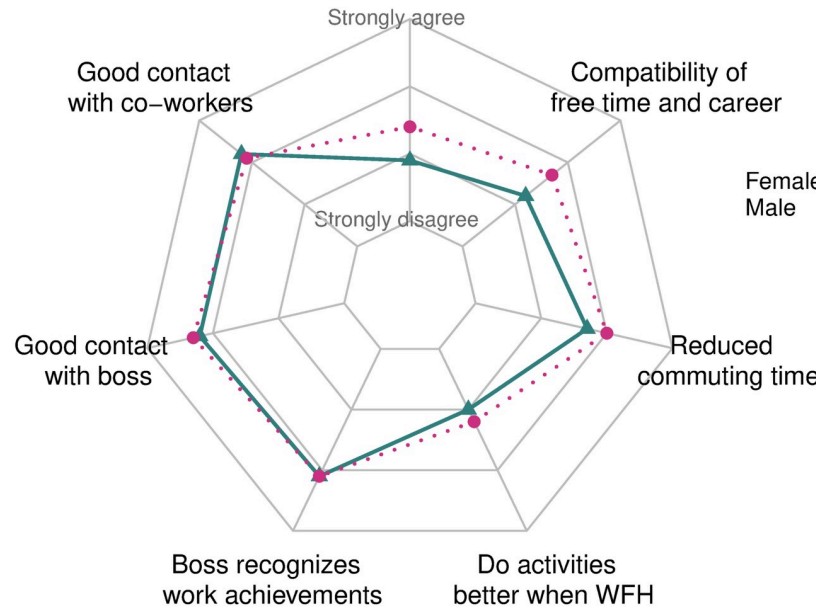

## (B) Quality of WFH

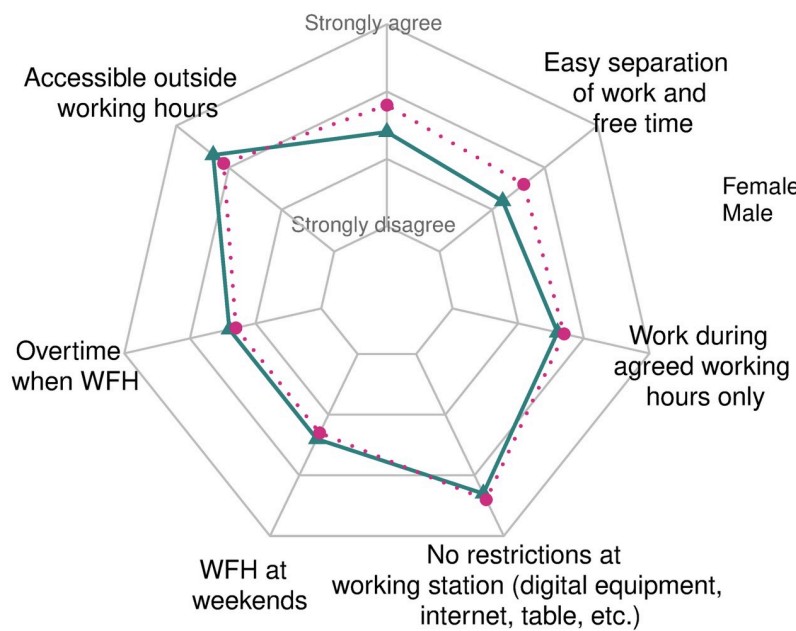

**Fig 1. Average agreement with statements on WFH from couple households with children.** *Reading example*: This radar chart displays the average agreement with different statements on WFH. The smaller the distance on the axis to the centre, the more the respondents disagree with the statement. Blue triangles represent answers from women, pink circles represent men's responses. On average, men (pink area) found 'compatibility of free time and career' to be more true than women (blue area), for example.

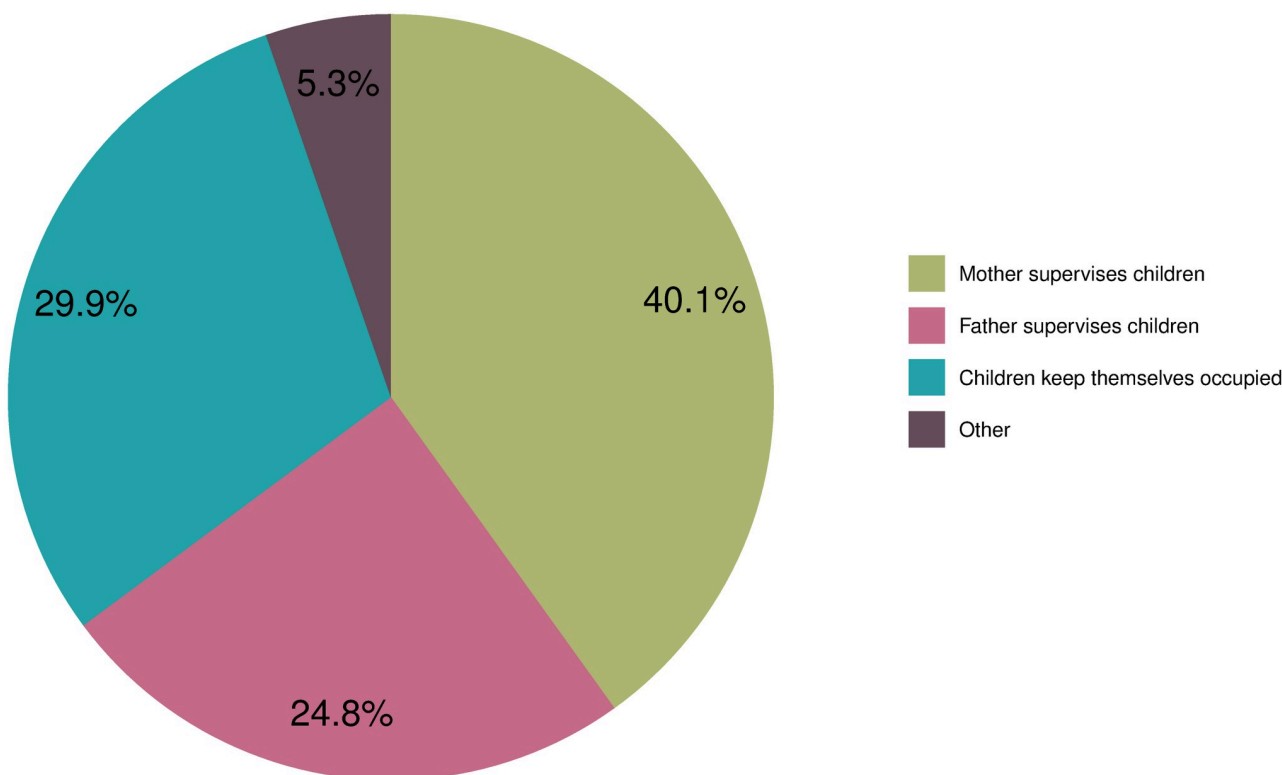

**Fig 2. Main childcare arrangement during working hours.** *Reading example*: The first green area indicates the share of respondents who stated that the mother supervises the children in the same room while WFH (40.1%). In contrast, only 24.8% of all respondents report that the father looks after the children and works simultaneously, while 29.9% of the children occupy themselves.

cases, the children took care of themselves, most probably because they were old enough to occupy themselves or do home schooling without much assistance. For 5% of the parents, external childcare was available, for example by relatives, friends, limited spaces in kindergartens or schools or some other form of paid care. These results might at least partially explain why women have more difficulties concentrating on their work compared to men, as they are more likely to supervise their children in the same room.

Overall, the results show that WFH is experienced differently by mothers and fathers. Since we only find minimal differences between the genders for households without children under 15 years old, we conclude that childcare is the most influential factor explaining difficulties in working from home. This is confirmed by an additional analysis, which shows that mothers find it more challenging than fathers to reconcile family and work and more commonly express feelings of guilt for neglecting their paid work and/or their children (see S2 Fig in S1 File). A likely explanation for this is the struggle for women to combine the demands of the professional world with their role as the primary caregiver, as gendered responsibilities still largely prevail in Austria.

**3.1.2 Division of unpaid work.** The conservative attitudes towards gender roles in Austria are also reflected in the unequal division of unpaid work. Missing information on time use before the pandemic prevents us from comparing absolute changes in hours spent on different activities before and during the COVID-19 lockdown. However, we asked respondents how they and their partner spent the previous working day during the lockdown (see S3 Table in S1 File). The results reveal that women in working couples spend, on average, almost two hours

more on unpaid work than men (4h03 compared to 5h58) per working day. The average time spent on unpaid work by women amounts to 6h44, compared to 7h48 for men. S3 Table in S1 File also includes and discusses information on differences in the participation rates (i.e. the share of respondents having spent some time on a certain activity) between men and women during the lockdown. Overall, we conclude that unusual times do not translate into unusual time use by gender.

Respondents were asked to evaluate the change in division of housework and childcare between both partners, before and during the stay-at-home orders. This enables us to analyse which partner primarily carried out which chores and whether the lockdown changed the division of work. Fig 3 reveals this division of unpaid work in couple households. The x-axis shows the division of unpaid work before the lockdown on a 11-point scale. The height of the bars indicate the number of couples identified with each scale number, divided into three groups. Each (coloured) group refers to changes of the division of unpaid work within couples, i.e. whether "the woman does more" (green bar), "nothing changed" (blue bar) or "the man does more" (grey-purple bar) during the lockdown.

Fig 3A and S3A Fig in S1 File reveal several aspects: first, Fig 3A shows that the distribution of housework is right-skewed on the 11-point scale. Prior to the lockdown, in 58% of all couple households women did the majority of housework (scale nos. 0–4), while it was equally divided in 28% of couples. In 15% of cases, the male partner was mainly responsible for doing housework before the outbreak of the pandemic. Hence, the data show that the division of housework complies with traditional gender roles in the majority of the observed couples. Men in couple households who spend more time on housework than their female partner are still an exception. Second, we examine the changes within couple households during the lockdown. The colours and the height of the bars shown in Fig 3A depict those changes. In almost half of all households (45%), the division of housework did not change (indicated by the blue bars). In 28% of all couples, women took on a larger share of housework during the lockdown than before (indicated by the green bars). The share of couples where men increased their share amounts to 27%. Two additional findings stand out in Fig 3A: first, in households where the division of housework was traditional before COVID-19 (scale nos. 1–3) and changed during

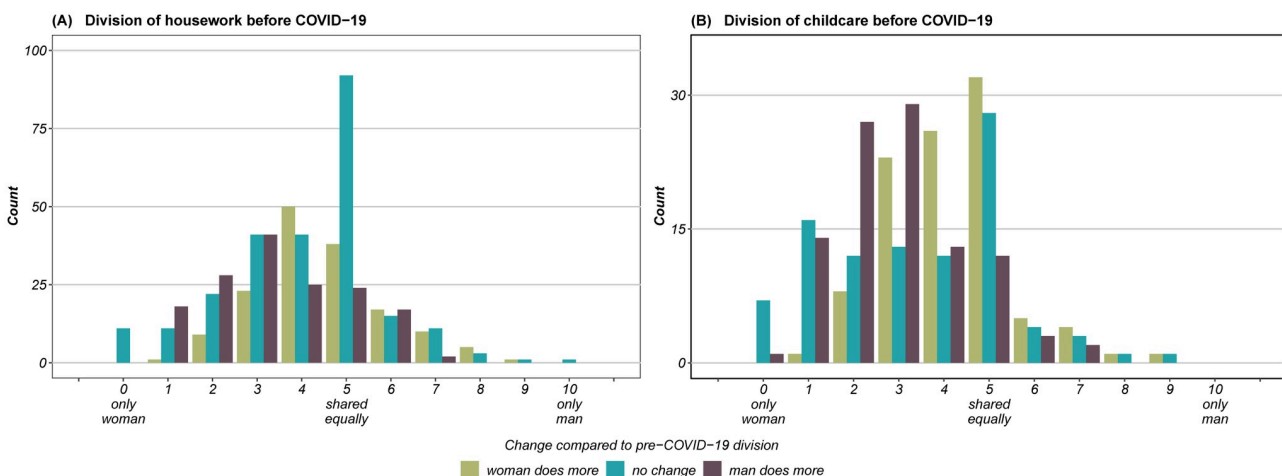

**Fig 3. Division of housework (A) and childcare (B) before COVID-19 and subsequent changes during lockdown.** *Reading example*: The histogram indicates the division of unpaid work before and according changes during the lockdown. The height of the bars state the number of couples divided into three coloured categories. Regarding housework (A): before the pandemic 152 couples shared housework equally (scale no. 5). For 92 of those couples nothing changed during the lockdown (blue bar). In 38 couples the female partner took on a larger share during the lockdown compared to before (green bar), whereas the opposite (male partner took on a larger share) holds true for the remaining 24 couples (grey-purple bar).

the stay-at-home orders, the division became more equal (i.e. men increased their share). Second, we observe a tendency towards retraditionalization of gender roles in households where housework was equally shared (scale no. 5) before the pandemic. Here, no change occurred in 60% of households that shared housework equally. For the remaining 40% of couples in which the division of housework did change (at scale no. 5), a retraditionalization (i.e. females doing now a larger share than before) occurred in almost two out of three households. Last, S3A Fig in S1 File shows the overall division of housework before and during the COVID-19 stay-at-home orders. In comparison to the division before COVID-19, the data reveal a slightly more polarized distribution. What stands out is that the number of households where the woman does everything roughly doubled from 11 (before) to 21 (during) cases. Nonetheless, the lockdown measures did not alter the overall distribution much.

The overall division of childcare is shown in S3B Fig in S1 File. Again, we observe a right-skewed distribution, indicating an unequal division of childcare. In comparison to housework, the division of childcare is more unequally divided. Before the COVID-19 restrictions, the main provider of childcare was women (68%). One in every four couples stated that childcare was equally shared between partners. Role reversal (i.e. fathers being the primary caregiver) is the exception (8%). During the pandemic, the distribution of childcare became slightly more polarized, but the overall distribution did not change significantly. Women still bore most of the childcare responsibilities, also during times of school closures. What is striking is that also for childcare, the amount of households where women were the sole caregiver (scale no. 0) almost doubled during the *lockdown* from 8 to 15. At the same time, there is no household that reports that the man does or did all the childcare. Regarding the specific within-couple changes see Fig 3B. We find that changes are more dynamic in the case of childcare compared to housework. The division remained unaltered in (32%) of couples. If changes occurred, these were again equally split between men doing more (34%) and women doing more (34%). The findings concerning changes to the division of housework also hold true for the division of childcare: men whose share of childcare was relatively low beforehand (scale nos. 1–3) mostly increased their share during the pandemic, whereas when childcare responsibilities were shared equally before COVID-19 (scale no. 5), a retraditionalization of gender roles can be observed. This is also the case when women did slightly more than 50/50 (scale no. 4) before the lockdown.

To summarize, we find that the pandemic did not substantially change the *overall* division of housework and childcare between men and women. In general it is still the case, that women bear on average more unpaid work than men. Nevertheless, for the majority of couples the lockdown had at least little effects on the division of unpaid labour *within* the household (55% for housework and 68% for childcare). The descriptive analysis further suggests that whether couples moved towards a more gender-equal division or not seems to depend strongly on the initial division of unpaid and paid work before the pandemic.

## 3.2 Regression results

The descriptive results presented in the previous subsection show how couple households divided the burden of unpaid work at the expense of women. In this section, we explain and discuss the change in the division of housework and childcare during as compared to before the lockdown. The main results of the three regression models are provided in Table 1. The section is structured as follows. We recap the specification of the dependent variables of the three distinct regression models and we describe the corresponding samples. Thereafter, we introduce and discuss the results of these models. This discussion starts with a debate of the main results of the paper. These are the distinct effect of WFH, the pre-lockdown division of

**Table 1. Average marginal effects of logistic regressions.**

| | *Dependent variable:* | | |
|---|---|---|---|
| | more HW: ♂ (1) | more HW: ♂ (2) | more CC: ♂ (3) |
| WFH: both | 0.15 (0.07) ** | 0.19 (0.10) ** | 0.11 (0.10) |
| WFH: only ♀ | 0.11 (0.10) | 0.06 (0.15) | -0.04 (0.12) |
| WFH: only ♂ | 0.23 (0.12) ** | 0.42 (0.14) *** | 0.30 (0.12) ** |
| WFH: nobody (= *ref*) | | | |
| HW before: ♀ more | 0.17 (0.05) *** | 0.15 (0.06) ** | |
| HW before: ♀ much more | 0.32 (0.06) *** | 0.33 (0.08) *** | |
| HW before: ♂ (much) more | 0.07 (0.07) | 0.14 (0.11) | |
| HW before: equal (= *ref*) | | | |
| CC before: ♀ more | | | 0.21 (0.06) *** |
| CC before: ♀ much more | | | 0.36 (0.07) *** |
| CC before: ♂ (much) more | | | 0.03 (0.12) |
| CC before: equal (= *ref*) | | | |
| Higher income: ♀ | 0.17 (0.07) ** | 0.21 (0.11) * | -0.01 (0.10) |
| Higher income: ♂ | 0.09 (0.04) ** | 0.11 (0.06) ** | 0.06 (0.06) |
| Equal income (= *ref*) | | | |
| Working hours ≤20h: ♀ | 0.01 (0.05) | -0.01 (0.06) | -0.11 (0.06) * |
| Working hours ≤20h (ST): ♀ | -0.02 (0.08) | 0.00 (0.09) | -0.03 (0.09) |
| Working hours >20h (ST): ♀ | -0.14 (0.15) | -0.18 (0.11) * | 0.05 (0.22) |
| Working hours >20h: ♀ (= *ref*) | | | |
| Working hours ≤20h: ♂ | 0.15 (0.10) | 0.34 (0.13) *** | 0.46 (0.09) *** |
| Working hours ≤20h (ST): ♂ | 0.06 (0.09) | 0.03 (0.10) | 0.14 (0.11) |
| Working hours >20h (ST): ♂ | 0.01 (0.10) | -0.19 (0.09) ** | -0.25 (0.10) ** |
| Working hours >20h: ♂ (= *ref*) | | | |
| Self-employed: ♀ | -0.06 (0.06) | -0.09 (0.08) | -0.20 (0.07) *** |
| Employed: ♀ (= *ref*) | | | |
| Self-employed: ♂ | -0.07 (0.05) | -0.04 (0.07) | -0.18 (0.07) *** |
| Employed: ♂ (= *ref*) | | | |
| No. children 0–2 years | -0.05 (0.05) | 0.03 (0.07) | 0.05 (0.07) |
| No. children 3–5 years | -0.06 (0.04) | -0.02 (0.05) | 0.05 (0.06) |
| No. children 6–9 years | 0.06 (0.03) * | 0.09 (0.04) * | 0.00 (0.05) |
| No. children 10–14 years | -0.06 (0.04) | -0.09 (0.06) | -0.17 (0.07) ** |
| Age: ♀ | -0.00 (0.00) | 0.01 (0.01) | 0.02 (0.01) ** |
| Age: ♂ | 0.00 (0.00) | 0.00 (0.01) | 0.00 (0.01) |
| Educ. ♀: Higher sec. | 0.02 (0.06) | -0.00 (0.08) | 0.09 (0.08) |
| Educ. ♀: Lower sec. \| prim. | -0.04 (0.07) | 0.01 (0.10) | 0.16 (0.11) |
| Educ. ♀: Tertiary (= *ref*) | | | |
| Educ. ♂: Higher sec. | 0.02 (0.05) | 0.00 (0.06) | 0.06 (0.06) |
| Educ. ♂: Lower sec. \| prim. | -0.00 (0.06) | -0.02 (0.09) | -0.09 (0.08) |
| Educ. ♂: Tertiary (= *ref*) | | | |
| Observations | 558 | 299 | 299 |
| Log likelihood | -299.55 | -148.15 | -152.85 |
| AIC | 599.09 | 296.30 | 305.70 |
| BIC | 653.09 | 350.30 | 359.70 |

*** $p < 0.01$;

** $p < 0.05$;

* $p < 0.1$

Note: WFH = working from home; HW = housework; CC = childcare; ST = short-time.

unpaid work, and the relative income of the partners living in a household that we employ as a measure for relative bargaining power. Next, we describe the most interesting findings in relation to other control variables included in these models. Importantly, we relate the results to the literature on gender bargaining, on the division of labour within household, and on task specialization within households. Overall, the findings stress the importance of studying the within household division of housework and childcare separately.

**Model (1)** explains the change in the division of housework. The binary dependent variable indicates whether the male partner took on (at least marginally) more housework than before. All heterosexual couples where both partners are either employed, self-employed or in short-time work and with full information on all covariates are included in the regression sample (h = 558).

**Model (2)** checks whether the effects of the explanatory variables on the probability that a man took on more housework during the lockdown are different for couples with children younger than 15. The dependent variable is the same as in model 1, however, the sample is different as couple households without children under age 15 are excluded (h = 299).

**Model (3)** explains the change in the division of childcare tasks (CC). The binary dependent variable equals one if the father took on (at least marginally) more childcare than before the restrictions. The sample is the same as in model 2 (h = 299).

**3.2.1 Working from home during lockdown.**   We find a positive and significant effect of the event that both partners were WFH during lockdown on the probability that a man took on a higher share of housework. In this case, the probability that a man increased his share in the conduct of housework is 15 percentage points higher than in the reference group, where nobody was WFH. With an increase of 23 percentage points, the effect of WFH is even larger when only the male partner worked from home. Moreover, the corresponding coefficients are also highly significant but even slightly larger in magnitude in the sub-sample that excludes households without at least one child younger than 15 years (see model (2) in Table 1). However, in both samples (model (1) and (2)), we do not find any significant effects on the probability of men doing relatively more housework than before in the event of only the female partner was WFH. Model (3) reports the results of the model explaining the change in the division of childcare tasks. Both parents WFH or solely the mother WFH have no significant effect on the probability of fathers increasing their share of childcare tasks. The effect of only a father WFH, by contrast, is significant and amounts to an increase in the probability by 30 percentage points, compared to the reference group.

There are two main results of the effect of WFH on the change in the division of unpaid work. First, we find a higher probability of men doing relatively more housework than before if both partners or only the male partner are WFH. Second, the effect is larger for childcare if fathers are WFH alone, but vanishes if mothers are also (or solely) WFH. The effects of both partners or only the male partner WFH on the division of housework are driven by households with children. As a robustness test (see model (4) in S4 Table in S1 File), we estimate a model based on the sample of childless couple households, and in this case, all effects of WFH on the division of housework turn insignificant. In other words, we do not find evidence that WFH influences the probability of men increasing their share of unpaid work within childless households. As long as mothers are at home, childcare seems to be mostly their responsibility, whereas fathers are more likely to take on more household chores instead. There are several potential factors that might drive this finding. In principle, it could be that those households initially had a more unequal pre-lockdown division of unpaid work. However, we control for

the pre-lockdown division of housework and childcare. The results may also be explained by a gendered specialization for certain household tasks [51–53]. Some studies [8, 54] show that men's share in grocery shopping increased during lockdown. The authors' explanation for the increase in time devoted to shopping by men is that this is an easy task, but a task that also carries a certain risk of infection. It is also conceivable that grocery shopping was welcomed by men to spend some time alone outside of their home. Our results can also be interpreted as a change in the task specialization by gender, to some extent. We find that both parents or only the mother WFH does not alter the probability of men taking on more childcare tasks, but it does have an impact on housework (if both partners are at home). This indicates that especially childcare is still strongly separated into traditional gender roles, even during (or rather also in) times of crisis.

**3.2.2 Pre-lockdown division of labour.** We find a significant and substantial effect of the pre-lockdown division of housework and childcare on the probability of men taking on a higher share of unpaid work during the COVID-19 restrictions. The variable capturing the pre-lockdown division of labour is a categorical variable, derived from a ranking of the female/male share of housework (HW) or childcare (CC) responsibilities as described in section 2.3. We include the pre-lockdown division of HW only in the regression explaining the change in the division of HW (model (1) and (2)), and the pre-lockdown division of CC only in the regression explaining the change in the division of CC (model (3)). Men and women who indicate an equal division of tasks serve as the reference group in the regressions. We find a significant effect of women being primarily responsible for HW and CC before the lockdown on the probability that the male partner does more HW and CC during the restrictions. In terms of their magnitude, these effects are similar to the effect of WFH. However, the effect is larger for couples where the woman took on (i) *much more* unpaid work than her partner, compared to households where the woman did just (ii) *moderately more* housework and childcare tasks. This finding indicates that it is relatively "easier" for men to do at least a little bit more of unpaid work during the lockdown restrictions when they initially fulfilled none or only a few tasks. The results hold true for all definitions of the sample (i.e. all couple households, couple households with children younger than 15 years, couple households without children). We hence conclude that these pre-lockdown division effects stem from households with and without children. However, we find no significant effect on men taking on more housework if they already took on a higher share of unpaid work before the restrictions, which in general are rare observations. Overall, these results show that changes in the division of unpaid work during the lockdown are largely influenced by the pre-lockdown division of HW and CC, and that change in times of crisis is easiest for men in couples in which the woman formerly did most of the unpaid work herself.

**3.2.3 Monthly net income.** We find mixed support for the predictions of gender bargaining models. In particular, we find no evidence of female bargaining power over the division of childcare. Bargaining models, especially the separate spheres approach [20], assume that the division of labour in couple households is the result of negotiations between the partners. Following this argument, the individual income of each partner represents a power resource that influences the division of labour. Therefore, the partner with the higher individual income and thus the higher share in total household income has more bargaining power and is able to influence the division of HW and CC in his/her interest. To control for such a mechanism, we include the relative income of the partners in the regression. The corresponding variable is a categorical variable, with three categories: both partners have equal income (reference group), the female outearns the male, or the male outearns the female partner. We find a significant and positive effect of the female partner having a higher income on the probability of men doing more housework than before the lockdown. In case the female outearns her male

partner, the probability of men doing more housework than before the lockdown is 17 percentage points higher as compared to the reference group, characterized by an equal contribution of the household's income. This roughly also holds true for the subsamples consisting only of households with and without children. These results suggest bargaining power as an underlying mechanism: if women earn more than their partners, their respective power (represented by income) transfers onto other fields of negotiation as well, such as division of housework. However, we do not find a positive effect of higher female income on the change in the division of CC. On top of that, we find that if men earn more than their female partners, the probability of men doing more housework rises (significantly) as well. At first, this may seem to be diametrically opposed to the theoretical prediction of bargaining power models, arguing that men should rather be doing less or the same amount of unpaid work if they hold more power (i.e. income). We explain this contradictory finding by the fact that households in which men outearn women are for the most part couples whose pre-lockdown division of unpaid work was already very unequal.

**3.2.4 Working hours.** Time-availability approaches [18] argue that couples face time pressure, and the partner spending fewer hours on paid labour will thus spend more time on housework. In model (1) we find no significant effect of working hours on the probability that the male partner takes on more unpaid work. A separate regression, where we reduced the sample to childless couples (see again model (4) in S4 Table in S1 File), shows that this result stems from households without children. Based on the theory of time availability, we expected to find that male partners working fewer hours for pay would have a positive likelihood of taking on more unpaid work as they have more free time. A differentiated picture emerges in models (2) and (3), based on households with children younger than 15 years only. Male part-time work shows a positive and significant effect on the change in the division of HW and CC, while male short-time full-time indicates a negative and significant effect, and male short-time part-time is insignificant. Stated differently, we find a positive effect of fewer working hours for voluntary part-time fathers, but a negative effect for fathers who were forced to work fewer hours in their full-time positions. In terms of their magnitude, the effect of fathers working part-time voluntarily on the probability to increase their share in housework and childcare are substantial. The positive effect of a man working part-time voluntarily might be driven by male selection into part-time. Those men might do so because they are willing to be actively involved in HW and/or CC. As the lockdown increased the burden of unpaid work, male involvement increased in response in these cases. With respect to the negative effect of male short-time full-time work (as compared to the reference group of full-time workers), the result can be interpreted as follows: in this group of workers, full-time work corresponds to any hours worked above 20 hours a week. As the lockdown increased the volume of unpaid work to be done within households, short-time full-time workers rather continued their role as primary earners while mothers continued their role as primary caregivers, which, under an overall increase of unpaid work, might imply that the share of unpaid work done by short-time full-time workers even decreases. This means that a change in working hours does not necessarily imply a change in involvement in unpaid work for this particular group. This is also reflected in the largely insignificant results on female working hours. Put differently, gender roles regarding the division of unpaid work do not automatically change due to fewer working hours. It seems that the majority of men do as much unpaid work as before the lockdown, conditional on their hours of paid work. The only exception is fathers voluntarily working part-time. As suggested by the gender display approach [21], norms play an important role in determining the division of work—also during crises.

**3.2.5 Number of children.** The number of children in different age categories seems to have an equivocal effect on the probability of men taking on more housework or childcare.

Model (1) compares all couple households, regardless of the number and age of children. In this case, we assigned childless households zero children in each age group. In model (1), we find no significant effect of an additional child in any age group compared to no (or fewer) children within the same age group. In model (2) we find a weakly significant and positive effect on men taking on more HW with each (additional) child between 6 and 9 years of age, while in model (3) we find a negative effect on men taking on more CC with each (additional) child between 10 and 14 years of age. Even though the effects are weak and small in magnitude, we interpret this to mean that children between 6 and 9 years old might represent a special age group, as they need more attention and support regarding homeschooling than younger or older age groups. Having more children between 6 and 9 years therefore means even more workload during lockdown, such that the probability of fathers doing more housework increases, probably leaving the childcare to the mothers. Older children might be more likely to manage the additional workload (e.g. homeschooling) themselves or even help their younger siblings. Overall, the absence of a clear and significant pattern points to the importance of persisting gender norms during the lockdown. The pre-lockdown division of unpaid work already depends (implicitly or explicitly) on the number of children within the household. That is to say, once we control for the pre-lockdown division of unpaid work, there remains no separate effect caused by the number and age of children.

**3.2.6 Additional control variables.**   The explanatory power of the additional control variables, the age of the female and male partners, their highest level of education completed and their employment status in terms of being either employed (reference category) or self-employed is limited. In this paragraph, however, we summarize the most important findings on age, education, and employment-status effects. In model (1) we find no significant effect of the employment status on the probability of men taking on more HW than before the lockdown. Analyzing the effect of the employment status on HW in separate samples of couple households with and without children (see model (2) and model (4) in S4 Table in S1 File), we do not find an effect among parent households, but a small and significant negative effect of male self-employment in the subsample of households without children. With respect to the change in the division of CC in model (3), both male and female self-employment has a negative and significant effect on the probability of fathers taking on more CC tasks. Both being negative suggests that different factors might be at play. For instance, the result might be driven by self-selection into self-employment based on the division of CC. Being self-employed frequently entails more flexibility, autonomy and the possibility to WFH, which facilitates reconciliation of work and family. This appears to be one reason why women with dependent children are more likely to be self-employed [55, 56]. Thus, self-selection of mothers into self-employment for family reasons might explain the negative and significant effect of their partners being less likely to increase their share in childcare activities. We only find a small positive and significant effect of female age on the probability that fathers take on more childcare, but it does not play a significant role for the change in the probability that a men took on more housework. Finally, there is no significant effect of education on the probability of men taking on more unpaid work during lockdown restrictions. Characteristics and structures defining the pattern of change in the share of unpaid work carried out by males are embodied in other variables, such as income or WFH. We conclude that these are factors that explain the independent variable (of men doing more unpaid work) better than education categories, ceteris paribus.

## 3.3 Robustness tests

We provide an extensive set of robustness tests in the S1 File. These checks alter the definition of the sample, control for the gender of the respondents, and are based on different

specifications of the control variables and the dependent variable. In addition, we present the results of the main models specified as linear probability models and estimated by ordinary least squares. Stressing the most important findings of these tests, first we find that the change in the probability of a male taking on more housework is driven by couple households with children. Second, different specifications of the income, age, age of the youngest child, and working hours variables do not alter the results, while the cutoff for classifying individuals as working either part-time or full-time matters. Third, overall these tests do not alter the results presented in the main text in any unexpected way. Hence we conclude that the main results are robust.

## 4 Limitations

First, we stress that it is necessary to interpret the findings in the context of the specific circumstances of the lockdown, specifically, the shift towards WFH in combination with the closure of childcare facilities and schools. It remains an open question how couples would have allocated unpaid work and experienced WFH if childcare facilities and schools had been open. Second, the average marginal effects of the main variables of interest (WFH and the pre-lockdown division of housework and childcare) on the change of within-household division are robust to different specifications and both statistically significant and large in magnitude. However, the standard errors of most of the control variables (the number of children per age group, age, education and employment status of both partners) are large. We interpret this in the sense that these controls have no considerable additional effect on the change in the division of housework and childcare once WFH, the pre-lockdown division of unpaid work and the hours worked for pay are taken into account. Third, we emphasize that the dependent variable depicts the change in the division of unpaid work, and not the change in hours spent on childcare and housework, respectively. An increase in male involvement in housework, may (and on average does) imply that women still spend more hours on unpaid housework. Finally, the sample is not representative of the Austrian working population. Compared to census data, it includes a disproportionately high number of individuals with a tertiary qualification. However, this is the group of couples that had to WFH more frequently than those with primary or secondary qualifications, who work more often in sectors considered "critical infrastructure". Thus, the sample stresses the change in the household division of unpaid work in the group of highly educated couples. As higher educational attainment is often associated with increased gender egalitarianism [57], we interpret the results rather as upper bounds of the involvement of males in HW and CC. The reason may be that constellations of *she* working part-time, and *he* being the primary earner might be more common among couples where neither partner has completed tertiary education [58].

## 5 Conclusion

In recent years, social scientists [48, 59, 60] have discussed the potential effects of WFH on the reconciliation of family and work. This debate was reopened following the COVID-19 pandemic that forced many individuals to WFH. While some argued that men would increase their share of unpaid work during the pandemic, others argued that gender roles and the gendered division of labour would intensify. To the best of our knowledge, we present the first study that closely examines the gendered aspects of the COVID-19 crisis in the overlapping spheres of paid and unpaid work and that explains the (change in the) division of unpaid work in couple households as a result of WFH. While pre-COVID-19 studies on the effects of WFH on the division of unpaid work suffered from selection bias, we have been able to investigate this effect by drawing on the very strict (and exogenous) lockdown. Even though the data

employed is not representative of the Austrian working population, it focuses on the parts of the population most likely to be able to WFH during the lockdown, and it contains rich information on the division of unpaid work and the experience of WFH. A key strength of this study is that it focuses on couples instead of individuals, thus offering unique insights into the division of work within households in relation to both partners' characteristics. In addition, while several studies documented the division of unpaid work during lockdowns, we are able to focus on how it *changed* during these extraordinary circumstances. This allows us to test whether a retraditionalization of gender roles can be observed in Austria, a country where conservative gender norms are predominant. We want to stress that the results show how couple households coped with the situation of WFH and unpaid work during the lockdown, but the findings cannot be transferred to WFH under "normal" conditions, with childcare institutions and schools being open.

The descriptive results reveal that unpaid work, especially childcare, has not been equally distributed within most couples either before or during the lockdown. The results from the econometric models indicate that men proportionally took on more housework during the lockdown than before in the event that both partners were WFH, or in the event that men were WFH alone (compared to those couples where nobody was WFH). Yet, this does not imply that men on average did more housework than their female partners in absolute terms, but simply that they took on a bigger share than before the COVID-19 crisis. While the econometric results do not provide information about how much more or which kind of housework was done by male partners, the descriptive results indicate that the steps towards a more equal distribution of unpaid work have been rather small. The descriptive analysis also shows that in households where the man's share of housework was very low before the lockdown, the division became (at least a little bit) more equal, whereas we observe a tendency towards a more traditional division of gender roles in households where housework was shared equally before the pandemic. This pattern is also confirmed in the econometric analysis. Furthermore, we do not find a significant effect of both partners WFH on the probability that fathers took on more childcare responsibilities than before the pandemic. This was only the case for couples where fathers were WFH alone. In addition, we find a significant effect of relative income differences on the probability that men take on more housework, whereas this does not seem to play a role for childcare. Strikingly, the results overall indicate that the division of childcare tasks is even more rigid than the division of housework. Given the massive increase in the volume of unpaid work due to the lockdown, one might have expected a more equal and even stronger involvement of males during these extraordinary circumstances.

WFH brings advantages and disadvantages for workers, but we find that they differ strongly by gender and household type. Working from home during the lockdown was very challenging, especially for mothers with children under 15 years. Mothers were more likely to find themselves stressed, working overtime, at weekends and with blurred boundaries between work and family time. Fathers were more likely to state that their concentration at home was good and that they had their own room to work from. When couples without children were asked about their experiences with WFH, the gender gaps almost vanished: both halves of the couples regard WFH as average equally good or bad. These findings are also reflected in the results of mothers feeling guilty for neglecting both their children and paid work.

We rather confirm than reject the notion that gender roles prevail during unusual times. The division of responsibilities for childcare tasks and the right "to work undisturbed" is not divided equally within couples. Primarily mothers had to watch the children during their working time, with fathers more often being able to rely on their partners doing that. WFH should therefore neither be regarded as a promising and automatic instrument to improve the reconciliation between family and career, nor as a way to promote more gender equity. Even

though it might help some families to better reconcile childcare and work, WFH during times where childcare facilities are closed puts more burden on mothers than fathers. As the sample focuses on well-educated, working couples in urban areas, the actual situation might even be more conservative and traditional than in this analysis. Therefore, the results should be regarded as "lower-bound" effects.

The findings, nevertheless, are a good reference point for policies that question current conceptions of work and that aim at promoting gender equality. Despite the data underlying this study being collected during unusual times, they provide valuable insights. WFH is often said to be a promising tool to improve the reconciliation between work and family life. As this study has shown, this does not hold true during hard lockdowns. Thus, the results further highlight the importance of the expansion of high-quality and affordable childcare facilities to ensure more gender equality at home and in the labour market. By providing institutional and publicly funded childcare, welfare states enhance gender equality, counteract dependencies within couples by facilitating full participation in the labour market and also improve the chances of children—especially from households that are economically worse off. Beyond the institutional setting, it is also worth highlighting the importance of promoting more equitable gender norms. Ideally this would start in kindergartens and schools, but should also expand to other settings, such as businesses, etc.

Finally, we can conclude that no automatic change comes out of crisis, and that nobody "lived happily ever after" without additional effort. In fact, we want to stress that *home* could be much *sweeter* for (working) mothers if they could rely on a more equitable division of unpaid work, especially in difficult times when the volume of unpaid work increases. That is to say, regarding the highly gendered specialization of tasks, in particular childcare, we need to include men, if we *all* want to be better off in the future.

## Supporting information

**S1 File. Additional figures, descriptive statistics, & robustness tests.**
(PDF)

## Acknowledgments

We thank numerous participants at various seminars and conferences, for instance, at the Vienna University of Economics and Business, the Research Institute Economics of Inequality, the Institute for Advanced Studies Vienna, the EFAS network, the Young Economists Conference 2020 and the Momentum Congress 2020 for their helpful comments and feedback on previous versions of this manuscript. In addition, we thank Wilfried Altzinger, Karin Heitzmann, Petra Sauer and Mathias Moser for comments on the manuscript. We thank Julia Hoffmann, Mathias Moser and Alyssa Schneebaum for commenting the questionnaire and the survey design.

## Author Contributions

**Conceptualization:** Judith Derndorfer, Franziska Disslbacher, Vanessa Lechinger, Katharina Mader, Eva Six.

**Data curation:** Judith Derndorfer, Franziska Disslbacher, Vanessa Lechinger, Eva Six.

**Formal analysis:** Judith Derndorfer, Franziska Disslbacher, Vanessa Lechinger, Eva Six.

**Funding acquisition:** Franziska Disslbacher, Katharina Mader.

**Investigation:** Franziska Disslbacher.

**Methodology:** Judith Derndorfer, Franziska Disslbacher, Vanessa Lechinger, Eva Six.

**Project administration:** Katharina Mader.

**Visualization:** Judith Derndorfer, Vanessa Lechinger, Eva Six.

**Writing – original draft:** Judith Derndorfer, Franziska Disslbacher, Vanessa Lechinger, Katharina Mader, Eva Six.

**Writing – review & editing:** Judith Derndorfer, Franziska Disslbacher, Vanessa Lechinger, Eva Six.

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
