## [Decision Letter · Decision Letter 0]

15 Jun 2021

PONE-D-21-13534

Home, sweet home? The impact of working from home on the division of unpaid work during the COVID-19 lockdown

PLOS ONE

Dear Dr. Disslbacher,

Thank you for submitting your manuscript to PLOS ONE. I read it with much interest and the reviewer also enjoyed your paper.

After careful consideration, we feel that it has merit but does not fully meet PLOS ONE’s publication criteria as it currently stands. Therefore, we invite you to submit a revised version of the manuscript that addresses the points raised during the review process.

In addition to reviewer comments, I have some observations.

1. Since your rely on multiple methods to collect interview data (mailing list, twitter, facebook), how does that impact your response rate and sample balance? In Austria, is there any gender difference in twitter and facebook usage? If so, could that affect gender balance in your original sample (N=2,113)? Please add a table in providing the breakdown of data by medium of interview/response and for each, provide % female.

2. Equally, is there similar difference by marital status? If so, did that affect the mix of single individuals and heterosexual couples in your original sample? I also wonder how your data collection method affects your answers given that individuals WfH would be more respective to digital communication and if so whether that inflates the proportion WfH in your data (owing to sample selection bias, since on page 8 you admit the WfH wasn't universal despite govt regulation)?

Again, I am sympathetic to "snowball sampling design" during COVID times. But we'd be open and frank about all the limitations including any systematic bias in sample composition and wherever possible, control for it in regression analysis (akin to "enumerator fixed effects" in face-to-face survey). Alternatively just acknowledge in your data limitation section. 

3. Please describe sample size more clearly. You claim to focus on "730 heterosexual couples (1,460 individuals)" but in table S1, it is adding up to 1377 (687+690) while in Table 1, it is 1159. Final sample size is expected to be stable across all results table. If you did not restrict analysis to a common sample (with non-missing cases), please revise all Tables accordingly. 

We look forward to receiving your revised manuscript.

Kind regards,

M Niaz Asadullah

Academic Editor

PLOS ONE

Journal Requirements:

2. Acknowledgments Section: Move New Information to the Financial Disclosure:

"Thank you for stating the following in the Acknowledgments Section of your manuscript: 

[copy in statement]

 Judith Derndorfer, Vanessa Lechinger, Katharina Mader and Eva Six thankfully acknowledge funding from the Vienna Science and Technology Fund (WWTF), grant number COV20-040, and the Chamber of Labour Vienna

grant Multiple Burdens of COVID-19. The funders had no role in study

design, data collection and analysis, decision to publish, or preparation of

the manuscript

4.Your ethics statement should only appear in the Methods section of your manuscript. If your ethics statement is written in any section besides the Methods, please delete it from any other section. 

5. You can choose to upload a striking image in Editorial Manager when you submit your manuscript. The image must be derived from a figure or supporting information file from your submission. To upload a striking image use the drop down menu on the “Attach Files” page to select “Striking Image” then select the image you would like to represent your manuscript. The striking image will not appear in the PDF sent to reviewers and editors, so it is important to make sure all necessary figures for the review process are uploaded as separate "Figure" file types.

Once your manuscript is accepted for publication, this image file will represent your article on the PLOS ONE homepage.

6. Your striking image file will represent your article upon publication on the PLOS ONE homepage. The image must be derived from a figure or supporting information file from your manuscript. Ideally, striking images should be high resolution, eye-catching, single panel images that do no contain additional text, scale bars, or arrows. 

Please also keep in mind that PLOS's Creative Commons Attribution License applies to striking images. As such, please do not submit any figures or photos that have been previously copyrighted unless you have express written permission from the copyright holder to publish under the CCAL license. You can read more about PLOS’s Creative Commons License on our homepage: http://journals.plos.org/plosone/s/licenses-and-copyright 

Reviewers' comments:

Reviewer's Responses to Questions

**Comments to the Author**

1. Is the manuscript technically sound, and do the data support the conclusions?

Reviewer #1: Partly

2. Has the statistical analysis been performed appropriately and rigorously? 

Reviewer #1: Yes

3. Have the authors made all data underlying the findings in their manuscript fully available?

Reviewer #1: Yes

4. Is the manuscript presented in an intelligible fashion and written in standard English?

Reviewer #1: Yes

5. Review Comments to the Author

Reviewer #1: The paper use the strict lockdown in Austria (which has conservative gender views) to explore the impact of working from home on the gender division of labour, arguing that the lockdown provides a natural experiment. The authors collected their own data through a survey and are very upfront about the limitations of the sample and stress the results are an upper bound. The paper finds that there was not much change in the division of labour and the main mechanism was child care. I think the paper is interesting and the results are of wide interest but I think the paper would need quite a bit of work before it meets the standards for publication, as follows:

1) The authors could provide a better sense of how much more unpaid work women do (globally and specifically to Austria/similar countries) rather than just stating women do more unpaid work

2) Firstly I would like to see much more discussion of the gender bargaining literature, how this paper fits in and a clear theoretical framework

3) I feel it was admirable to collect time use data but I felt these were under-utilised

4) I completely understand the reasons behind not being able to see the change in hours but I find this a big limitation - is there any way to combine the time use data and who does more to get sense of how much the men who do more are really doing? Men could double their unpaid work but if this is from 1 hour to 2 hours this is not much of a change...

5) Do we know how many of the respondents worked from home before the lockdown or had the option to?

6)I'm not sure I fully understand how pre-lockdown division is captured

7)The majority of responses come from women, have you used only those coming from the women as a robustness check i.e excluding those coming from men/both and compared responses when you have information from both partners? I imagine that there may be some disagreement on how does how much between partners (I have seen this in other datasets where division of labour is collected from both partners)

8) I found figures 1-3 quite difficult to interpret, I wonder if there are better ways to represent the descriptives, and draw out the key descriptive results?

9) I would like the discussion of the results to consider more of the magnitude of results and to have a greater understanding of how these results fit into the gender bargaining and division of labour literature. I think the initial discussion is there but could benefit from reference to more literature and theory

10) Would it also help to use the age of the youngest child instead of the number of children as a robustness check - I wonder how important the age of children are in women doing more of the child care? Especially having very young children?

11) I wonder if grocery shopping was appealing as it was an excuse to get out of the house!

12) Note the links to the supplementary material did not work for me so I could not see this

13) I found it quite hard to get to the key results so the authors may want to consider how to restructure the results and discussion to guide the reader to the key findings

6. PLOS authors have the option to publish the peer review history of their article (what does this mean?). If published, this will include your full peer review and any attached files.

Reviewer #1: No

---

## [Author Response · Author response to Decision Letter 0]

5 Aug 2021

We provide detailed responses to the specific reviewer and editor comments in the attached PDF file (responses.pdf).

---

## [Decision Letter · Decision Letter 1]

22 Oct 2021

Home, sweet home? The impact of working from home on the division of unpaid work during the COVID-19 lockdown

PONE-D-21-13534R1

Dear Dr. Disslbacher,

We’re pleased to inform you that your manuscript has been judged scientifically suitable for publication and will be formally accepted for publication once it meets all outstanding technical requirements.

Kind regards,

M Niaz Asadullah

Academic Editor

PLOS ONE

Additional Editor Comments (optional):

Reviewers' comments:

Reviewer's Responses to Questions

**Comments to the Author**

1. If the authors have adequately addressed your comments raised in a previous round of review and you feel that this manuscript is now acceptable for publication, you may indicate that here to bypass the “Comments to the Author” section, enter your conflict of interest statement in the “Confidential to Editor” section, and submit your "Accept" recommendation.

Reviewer #1: All comments have been addressed

2. Is the manuscript technically sound, and do the data support the conclusions?

Reviewer #1: Yes

3. Has the statistical analysis been performed appropriately and rigorously? 

Reviewer #1: Yes

4. Have the authors made all data underlying the findings in their manuscript fully available?

Reviewer #1: Yes

5. Is the manuscript presented in an intelligible fashion and written in standard English?

Reviewer #1: Yes

6. Review Comments to the Author

Reviewer #1: (No Response)

7. PLOS authors have the option to publish the peer review history of their article (what does this mean?). If published, this will include your full peer review and any attached files.

Reviewer #1: No

---

## [Editor Report · Acceptance letter]

29 Oct 2021

PONE-D-21-13534R1 

Home, sweet home? The impact of working from home on the division of unpaid work during the COVID-19 lockdown 

Dear Dr. Disslbacher:

I'm pleased to inform you that your manuscript has been deemed suitable for publication in PLOS ONE. Congratulations! Your manuscript is now with our production department. 

Kind regards, 

on behalf of

Dr. M Niaz Asadullah 

Academic Editor

PLOS ONE